# Construction of a CQDs/Ag_3_PO_4_/BiPO_4_ Heterostructure Photocatalyst with Enhanced Photocatalytic Degradation of Rhodamine B under Simulated Solar Irradiation

**DOI:** 10.3390/mi10090557

**Published:** 2019-08-23

**Authors:** Huajing Gao, Chengxiang Zheng, Hua Yang, Xiaowei Niu, Shifa Wang

**Affiliations:** 1State Key Laboratory of Advanced Processing and Recycling of Non-Ferrous Metals, Lanzhou University of Technology, Lanzhou 730050, China; 2Chongqing Key Laboratory of Geological Environment Monitoring and Disaster Early-Warning in Three Gorges Reservoir Area, Chongqing Three Gorges University, Chongqing 404000, China; 3School of Electronic and Information Engineering, Chongqing Three Gorges University, Chongqing 404000, China

**Keywords:** carbon quantum dots, CQDs/Ag_3_PO_4_/BiPO_4_, photodegradation activity, synergistic effect, photocatalytic mechanism

## Abstract

A carbon quantum dot (CQDs)/Ag_3_PO_4_/BiPO_4_ heterostructure photocatalyst was constructed by a simple hydrothermal synthesis method. The as-prepared CQDs/Ag_3_PO_4_/BiPO_4_ photocatalyst has been characterized in detail by X-ray diffraction, field-emission scanning electron microscopy, transmission electron microscopy, X-ray photoelectron spectroscopy, ultraviolet–visible spectroscopy, and photoelectrochemical measurements. It is demonstrated that the CQDs/Ag_3_PO_4_/BiPO_4_ composite is constructed by assembling Ag_3_PO_4_ fine particles and CQDs on the surface of rice-like BiPO_4_ granules. The CQDs/Ag_3_PO_4_/BiPO_4_ heterostructure photocatalyst exhibits a higher photocatalytic activity for the degradation of the rhodamine B dye than that of Ag_3_PO_4_, BiPO_4_, and Ag_3_PO_4_/BiPO_4_. The synergistic effects of light absorption capacity, band edge position, separation, and utilization efficiency of photogenerated carriers play the key role for the enhanced photodegradation of the rhodamine B dye.

## 1. Introduction

The photocatalytic degradation of organic pollutants in wastewater is an attractive, environmentally friendly and green method that offers a way to harness solar power efficiently and convert them into non-toxic degradation products [1,2,3,4,5,6,7,8]. Recently, although great progress has been made in the field of photocatalysis, only few photocatalysts can effectively use visible light in the degradation of organic pollutions. Therefore, it is desirable to develop novel photocatalysts with high visible-light utilization for degradation of organic pollutions in wastewater. In recent years, silver phosphate (Ag_3_PO_4_) based composite photocatalysts, such as Bi_4_Ti_3_O_12_/Ag_3_PO_4_ [9], Ag_3_PO_4_/NaTaO_3_ [10], MoS_2_/Ag_2_S/Ag_3_PO_4_ [11], Ag_3_PO_4_/Bi_2_WO_6_ [12], Ag_3_PO_4_/Cu_2_O [13], TiO_2_/Ag_3_PO_4_/bentonite [14], Co_3_(PO_4_)_2_/Ag_3_PO_4_ [15], and Ag_3_PO_4_/BiFeO_3_ [16] have been extensively studied due to their excellent photocatalytic activity for photocatalytic degradation of the organic pollutions under visible light irradiation.

Bismuth phosphate (BiPO_4_) as a photocatalyst has been widely studied because of its good photoelectric performance, low cost, low toxicity, excellent photocatalytic activity, and high stability [1]. However, the large optical bandgap (*E*_g_ = 4.5 eV) of BiPO_4_ limits the transmission efficiency of photon-generated carriers and light-response range to sunlight [17,18]. To expand the photoresponse range of BiPO_4_, constructing composite photocatalysts with Ag_3_PO_4_ (*E*_g_ = 2.43 eV) can effectively improve the photocatalytic activity of the composite photocatalysts [19,20,21,22,23,24]. However, the Ag_3_PO_4_/BiPO_4_ photocatalysts have a high recombination rate of photogenerated electrons (e^−^) and holes (h^+^) in the degradation of organic pollutions in wastewater [25]. To achieve excellent photocatalytic performances of the photocatalysts, the photoexcited electrons and holes must be efficiently separated [26,27,28,29].

Noble metal nanoparticles (NPs) and carbon nanomaterials including carbon quantum dots (CQDs), carbon nanotubes (CNTs) and graphene manifest many intriguing physicochemical characteristics and offer a wide scope of technological applications in electronic devices, biomedicine, sensors, and wave absorption [30,31,32,33,34,35,36,37]. These nanomaterials are good carrier transport materials and also exhibit interesting localized surface plasmon resonance (LSPR) effect or photoluminescence (PL) up-conversion effect [38,39,40]. Due to these outstanding properties, noble metal NPs, CQDs, CNTs, and graphene have been demonstrated to be excellent modifiers or co-catalysts to enhance the photocatalytic performances of semiconductor photocatalysts [41,42,43,44,45,46].

Herein, we report a hydrothermal synthesis of unique CQDs/Ag_3_PO_4_/BiPO_4_ heterostructure photocatalyst. The composite photocatalyst with the CQDs, Ag_3_PO_4_, and BiPO_4_ three phase junction structure has not been reported previously and may be commonly applicable to other composite photocatalyst systems. The as-obtained CQDs/Ag_3_PO_4_/BiPO_4_ heterostructure photocatalyst possesses a high light absorption capacity, high utilization and separation efficiency of photogenerated carriers, and exhibits a high photocatalytic activity for photocatalytic degradation of the rhodamine B (RhB) dye. The present CQDs/Ag_3_PO_4_/BiPO_4_ heterostructure photocatalysts can be used for the design of micro/nano-photocatalytic devices for the wastewater treatment.

## 2. Materials and Methods

### 2.1. Synthesis of the Ag_3_PO_4_ Photocatalyst

According to the formula Ag_3_PO_4_, an amount of silver nitrate (AgNO_3_) was mixed with a stoichiometric amount of sodium dihydrogen phosphate (NaH_2_PO_4_) powder with Ag/P = 3:1 and added into 60 mL distilled water. After that, a stoichiometric amount of ammonium hydroxide (NH_3_·H_2_O) was added to the mixture. The whole process was accompanied by magnetic stirring. Subsequently, the above mixture was transferred to a 100 mL high-pressure autoclave and heated to 160 °C for 6 h. After finishing the hydrothermal reaction, the content was taken out and washed with distilled water several times to remove excess alkali ions. The slurry was centrifuged and dried for 12 h at 80 °C to obtain the Ag_3_PO_4_ photocatalyst. The flow-chart for the synthesis of Ag_3_PO_4_ photocatalyst via the hydrothermal synthesis method is shown schematically in Figure 1(I).

### 2.2. Synthesis of the BiPO_4_ Photocatalyst

According to the formula BiPO_4_, stoichiometric amounts of bismuth nitrate pentahydrate (Bi(NO_3_)_3_·5H_2_O), NaH_2_PO_4_ and NH_3_·H_2_O were successively added in 20 mL of dilute nitric acid solution (2 mL HNO_3_ + 18 mL distilled water). The role of HNO_3_ is to dissolve Bi(NO_3_)_3_·5H_2_O. The mixture was filled up to 60 mL by adding distilled water. The remaining experimental steps are consistent with Section 2.1. The flow-chart for the synthesis of BiPO_4_ photocatalyst via the hydrothermal method is shown schematically in Figure 1(II).

### 2.3. Synthesis of Ag_3_PO_4_/BiPO_4_ Photocatalyst

To prepare Ag_3_PO_4_/BiPO_4_ photocatalyst, stoichiometric amounts of Bi(NO_3_)_3_·5H_2_O, AgNO_3_, NaH_2_PO_4_ and NH_3_·H_2_O (*n*_Ag3PO4_:*n*_BiPO4_ = 1:0.11) were successively added in 20 mL of dilute HNO_3_ solution, and then filled up to 60 mL by adding distilled water. The assembly of Ag_3_PO_4_ on BiPO_4_ followed the procedure as described in Section 2.1. The flow-chart for the synthesis of the Ag_3_PO_4_/BiPO_4_ photocatalyst is schematically shown in Figure 1(III).

### 2.4. Synthesis of CQDs/Ag_3_PO_4_/BiPO_4_ Photocatalyst

To obtain the CQDs/Ag_3_PO_4_/BiPO_4_ photocatalyst, stoichiometric amounts of Bi(NO_3_)_3_·5H_2_O, AgNO_3_, NaH_2_PO_4_ and NH_3_·H_2_O and 6 mL of the CQDs suspension derived according the literature [45] were successively 20 mL of dilute HNO_3_ solution, and then filled up to 60 mL by adding distilled water. The subsequent preparation process is consistent with Section 2.1. The flow-chart for the synthesis of the CQDs/Ag_3_PO_4_/BiPO_4_ photocatalyst is shown in Figure 1(IV).

### 2.5. Sample Characterization

The phase purity of the Ag_3_PO_4_, BiPO_4_, Ag_3_PO_4_/BiPO_4_ and CQDs/Ag_3_PO_4_/BiPO_4_ photocatalysts was analyzed by means of D8 advanced X-ray diffractometer with Cu Kα radiation at a wavelength of 1.5406 Å. The surface morphology of the samples was characterized by JSM-6701F field-emission scanning electron microscopy (SEM, JEOL Ltd., Tokyo, Japan) and JEM-1200EX field-emission transmission electron microscopy (TEM, JEOL Ltd., Tokyo, Japan). Ultraviolet–visible (UV–VIS) diffuse reflectance spectra of the samples were examined on a UV–VIS spectrophotometer with an integrating sphere attachment using BaSO_4_ as the reference. To determine the bonding states, chemical composition, and electron levels of the samples, X-ray photoelectron spectroscopy (XPS) measurements were carried out by using a PHI-5702 X-ray photoelectron spectrometer (Physical Electronics, Hanhassen, MN, USA).

The electrochemical properties of the samples were investigated according to the method reported in the literature [45]. A CST 350 electrochemical workstation (Wuhan Corrtest Instruments Co., Ltd., Wuhan, China) equipped with a three-electrode cell configuration was used to study the electrochemical impedance spectroscopy (EIS) and photocurrent response of the samples. The working electrode was prepared as follows: 15 mg of the photocatalyst, 0.75 mg of polyvinylidene fluoride (PVDF), 0.75 mg of carbon black and 1 mL of 1-methyl-2-pyrrolidione (NMP) were mixed together to form uniform slurry. The slurry mixture was homogeneously coated on the surface of fluorine-doped tin oxide (FTO) thin film (effective area: 1 × 1 cm^2^), and subjected to drying 60 °C for 5 h. The used electrolyte was 0.1 mol L^−1^ Na_2_SO_4_ aqueous solution. The used light source was a 200 W xenon lamp emitting simulated sunlight. A 0.2 V bias voltage was used during the transient photocurrent measurement. The sinusoidal voltage pulse was used for the EIS measurement (amplitude: 5 mV; frequency range: 10^−2^–10^5^ Hz).

### 2.6. Photocatalytic Testing

The photocatalytic activities of the samples were investigated by removing RhB from aqueous solution according to the procedure as described in the literature [45]. A 200-W xenon lamp (sunlight simulator) was used as the light source. The photocatalytic system was composed of 0.1 g photocatalyst and 100 mL RhB solution (*C*_photocatalyst_ = 1 g L^−1^, *C*_RhB_ = 5 mg L^−1^). Based on the initial RhB concentration (*C*_0_) and residual RhB concentration (*C*_t_), the degradation percentage (DP) of RhB was given as: DP = (*C*_0_ − *C*_t_)/*C*_0_ × 100%.

## 3. Results and Discussion

### 3.1. Phase Structural Analysis

Figure 2a,b show the XRD patterns of Ag_3_PO_4_ and BiPO_4_, respectively. For the Ag_3_PO_4_ and BiPO_4_ samples, the XRD curves were fitted using the Jade 6.0 package. The black curves, red curves, vertical olive lines, and blue lines represent the observed XRD peaks, theoretically estimated curves, Bragg peaks, and difference between the observed values, and theoretically estimated values of XRD diffraction peaks, respectively. The result indicates that the theoretically simulated values are in good agreement with the observed XRD diffraction peaks. The XRD diffraction peaks of Ag_3_PO_4_ and BiPO_4_ can be ascribed to JCPDF#06-0505 and JCPDF#15-0767, respectively. Figure 2c shows the XRD patterns of Ag_3_PO_4_/BiPO_4_ and CQDs/Ag_3_PO_4_/BiPO_4_. The main XRD diffraction peaks of the Ag_3_PO_4_/BiPO_4_ and CQDs/Ag_3_PO_4_/BiPO_4_ composites are similar to those of pure Ag_3_PO_4_, indicating that the host lattice of Ag_3_PO_4_ in these composites undergoes no change. In addition to the XRD characteristic peaks of the Ag_3_PO_4_ phase, the XRD characteristic peaks of BiPO_4_ are also observed in these composites. For the CQDs/Ag_3_PO_4_/BiPO_4_ composite, the intensity of the diffraction peaks is sharper than that for Ag_3_PO_4_/BiPO_4_. The structure analysis shows that the introduction of CQDs in the Ag_3_PO_4_/BiPO_4_ composites obviously accelerate the formation of Ag_3_PO_4_ and BiPO_4_. In our previous study, the carbon can suppress the formation of M-ferrite [47] and α-Al_2_O_3_ [48] phase prepared by a polyacrylamide gel method. In this case, this phenomenon may be due to the fact that CQDs do not react with oxygen in the reactor to form carbon dioxide. Figure 2d,e show the crystal structures of BiPO_4_ and Ag_3_PO_4_, respectively. The BiPO_4_ and Ag_3_PO_4_ are monoclinic phase with space group P21/n (14) and cubic phase with space group P-43n (218), respectively. For the BiPO_4_, the Bi atom and the P atom are surrounded by eight oxygen atoms and four oxygen atoms, respectively. The wide Bi–O and P–O bond length of BiPO_4_ exhibits a high photocatalytic activity for photocatalytic degradation of organic pollutants [49]. For the Ag_3_PO_4_, the Ag atom, P atom and O atom experience four-fold coordination by four O atoms, four-fold coordination by four O atoms, and 4-fold coordination by one P atom and three Ag atoms, respectively [50].

### 3.2. Surface Morphology and Elemental Component Analysis

Figure 3a,b show the SEM images of Ag_3_PO_4_/BiPO_4_ and CQDs/Ag_3_PO_4_/BiPO_4_, respectively. For the Ag_3_PO_4_/BiPO_4_ composite, the sample is composed of fine spherical particles and rice-like granules, as shown in Figure 3a. Figure 3b represents the SEM image of the CQDs/Ag_3_PO_4_/BiPO_4_ composite, revealing that its morphology is very similar to that of the Ag_3_PO_4_/BiPO_4_ composite. The insets in Figure 3a,b show the real photos of Ag_3_PO_4_/BiPO_4_ and CQDs/Ag_3_PO_4_/BiPO_4_, respectively. The results show that the introduction of CQDs to the Ag_3_PO_4_/BiPO_4_ composite deepens the color of the sample. The detailed analysis will be done in the optical properties section. Figure 3c shows the SEM image of pure CQDs, from which it is seen that the prepared CQDs have a narrow size distribution of 7–10 nm.

The microstructure and elemental composition of the CQDs/Ag_3_PO_4_/BiPO_4_ composite was characterized by TEM, as shown in Figure 4. Figure 4a displays the TEM image of the composite. Spherical fine particles (Ag_3_PO_4_) are seen to be assembled on the surface of rice-like granules (BiPO_4_). The high-resolution TEM (HRTEM) image further confirms the assembly of Ag_3_PO_4_ fine particles on the surface of BiPO_4_ rice-like granules, as depicted in Figure 4b. The rice-like granules manifest obvious lattice fringes with an interlayer spacing of 0.347 nm, which correspond to the (222) facet of the cubic Ag_3_PO_4_ phase. The attached spherical particles exhibit the lattice fringes with a d-spacing of 0.407 nm, which correspond to the (101) facet of the monoclinic Ag_3_PO_4_ phase. The decorated ultrafine particles with no lattice fringes could be CQDs. The energy-dispersive X-ray spectroscopy (EDS) spectrum (Figure 4c) demonstrates that the elemental composition of the CQDs/Ag_3_PO_4_/BiPO_4_ composite is Ag, Bi, P, O, and C. Additional Cu signal observed on the EDS spectrum can be ascribed to the TEM microgrid holder [51]. To further elucidate the spatial distribution of elements, Figure 4b shows the dark-field scanning TEM (DF-STEM) image of the CQDs/Ag_3_PO_4_/BiPO_4_ composite and Figure 4e–i display the corresponding elemental maps. Ag, P, O, Bi, and C elementals are homogenously distributed throughout the rice-like granules, implying the uniform decoration of Ag_3_PO_4_ nanoparticles and CQDs on the surface of rice-like BiPO_4_ granules. The observed C element in the blank area without the sample could come from the TEM microgrid holder.

### 3.3. XPS Analysis

To understand the chemical composition and electronic core levels of the Ag_3_PO_4_/BiPO_4_ and CQDs/Ag_3_PO_4_/BiPO_4_ composites, Figure 5 shows the XPS results of the two composites. In Figure 5a, the XPS survey scan spectra for the Ag_3_PO_4_/BiPO_4_ and CQDs/Ag_3_PO_4_/BiPO_4_ composites clearly contain the P, Bi, Ag, O, and C elements. The electronic core levels of Bi 4f, P 2p, Ag 3d, O 1s, and C 1s in the composites are further characterized using the high-resolution XPS spectra. Figure 5b shows the Bi 4f core-level XPS spectra. Two obvious characteristic peaks at 161.02/160.29 and 166.26/165.63 eV are observed on the spectra, which are assigned to Bi 4f_7/2_ and Bi 4f_5/2_ binding energies of Bi^3+^ in BiPO_4_, respectively [52].

The XPS spectra of P 2p core level shown in Figure 5c present a broad peak at 134.36 (or 133.69) eV, suggesting that P species exhibits +5 oxidation state [52]. Figure 5d shows the Ag 3d core level XPS spectra. The peaks at 369.59/368.86 and 375.58/374.79 eV can be assigned to Ag 3d_5/2_ and Ag 3d_3/2_ of Ag_3_PO_4_, respectively [53]. For the O 1s core-level XPS spectra, the peak at 531.63/532.28 eV can be ascribed to the lattice oxygen, while the peak at 532.92/533.53 eV is related to the adsorbed oxygen [54,55], as shown in Figure 5e. The C 1s core-level XPS spectra are shown in Figure 5f. For the Ag_3_PO_4_/BiPO_4_ composite, the peak at 284.77 eV can be assigned to the adventitious hydrocarbon for the XPS instruments [56]. For the CQDs/Ag_3_PO_4_/BiPO_4_ composite, the C 1*s* peak can be divided in to three separate peaks at 283.75, 284.77 and 286.38 eV, corresponding to CQDs [57], adventitious hydrocarbon [56] and impurity structure of carbon [58]. It is noted that the electronic core levels of Bi 4f, P 2p, Ag 3d and O 1s for the CQDs/Ag_3_PO_4_/BiPO_4_ composite are smaller (about 0.61–0.73 eV) than those for the Ag_3_PO_4_/BiPO_4_ composite, which could be due to the fact that the CQDs facilitate the formation of CQDs/Ag_3_PO_4_/BiPO_4_ heterostructures.

### 3.4. Optical Properties

It is noted that the optical properties of semiconductors have an important effect on their photocatalytic performances, which can be determined by UV–vis DRS measurements [59]. Figure 6a shows the UV–VIS diffuse reflectance spectra of the Ag_3_PO_4_/BiPO_4_ and CQDs/Ag_3_PO_4_/BiPO_4_ photocatalysts. For both the samples, the reflectance first increases and then decreases with the increase in the wavelength, and finally increases again. The two samples present higher reflectance in the wavelength range from 550 to 850 nm. When CQDs are introduced to Ag_3_PO_4_/BiPO_4_, a decrease in the reflectance of the resultant CQDs/Ag_3_PO_4_/BiPO_4_ composite in the wavelength range from 300 to 850 nm is observed. According to the literatures [60], the color parameters (*L**, *a**, *b**), chroma parameter (*c**), hue angle (*H*^o^), and total color difference (*E*_CIE_*) of Ag_3_PO_4_/BiPO_4_ and CQDs/Ag_3_PO_4_/BiPO_4_ are evaluated, as shown in Table 1. The Ag_3_PO_4_/BiPO_4_ composite shows a negative a* value, indicating that it displays a reseda, as shown in the inset of Figure 3a. The CQDs/Ag_3_PO_4_/BiPO_4_ composite exhibits the smaller *L** and *b** values and positive *a** value, which means it exhibits yellowish black, as shown in the inset of Figure 3b. The first derivative curves of UV-vis diffuse reflectance spectra are useful to determine the optical bandgaps (*E*_g_) of semiconductors [61]. As shown in Figure 6b, the Ag_3_PO_4_/BiPO_4_ composite shows two absorption edges at 276.1 and 502.3 nm, whereas the CQDs/Ag_3_PO_4_/BiPO_4_ composite exhibits an absorption edge at 271.9 nm. The absorption edges at 276.1/271.9 and 502.3 nm can be assigned to BiPO_4_ and Ag_3_PO_4_, respectively. The disappearance of the Ag_3_PO_4_ absorption peak on the spectrum of the CQDs/Ag_3_PO_4_/BiPO_4_ composite is ascribed to the enhanced optical absorption caused by CQDs. The *E*_g_ values of Ag_3_PO_4_ and BiPO_4_ in the samples (see Table 1) can be derived on the basis of Equation (1):
(1)$$E_{g}(\mathrm{eV})=\frac{hc}{\lambda_{0}(\mathrm{nm})}\approx \frac{1240}{\lambda_{0}(\mathrm{nm})}$$
where *λ*_0_, *h*, and *c* is the maximum absorption wavelength, Plank constant, and velocity of light, respectively.

### 3.5. Photoelectrochemical Properties

Figure 7a shows the EIS spectra of the Ag_3_PO_4_/BiPO_4_ and CQDs/Ag_3_PO_4_/BiPO_4_ composites. For the two samples, the EIS spectra show a semicircle and a straight line, which can be ascribed to the charge transfer and the Warburg impedance, respectively [62,63]. The CQDs/Ag_3_PO_4_/BiPO_4_ photocatalyst has a smaller semicircle than that for the Ag_3_PO_4_/BiPO_4_ photocatalyst, which means the former exhibits a higher photocatalytic activity. Photocurrent response curves can also be used to predict the photocatalytic activity of semiconductor materials [64]. Figure 7b shows the photocurrent response curves of the Ag_3_PO_4_/BiPO_4_ and CQDs/Ag_3_PO_4_/BiPO_4_ photocatalysts. The photocurrent response of Ag_3_PO_4_/BiPO_4_ can be attributed to the electron transfer between Ag_3_PO_4_ and BiPO_4_. The CQDs/Ag_3_PO_4_/BiPO_4_ photocatalyst exhibits a higher photocurrent intensity than that of Ag_3_PO_4_/BiPO_4_, indicating that it possesses a higher photocatalytic activity because of its higher electron transfer and separation efficiency.

### 3.6. Photocatalytic Activity

To study the photocatalytic activity of the BiPO_4_, Ag_3_PO_4_, Ag_3_PO_4_/BiPO_4_, and CQDs/Ag_3_PO_4_/BiPO_4_ photocatalysts, RhB dye was used as a degradation dye. Figure 8a shows the time-dependent photodegradation of RhB in the presence of the samples under simulated sunlight irradiation. Based on the blank experiment, the RhB dye exhibits a high stability and is non-biodegradable at ambient conditions. The dye degradation rate over the samples increases with increasing the irradiation time. The photocatalytic activity of these photocatalysts follows the order: CQDs/Ag_3_PO_4_/BiPO_4_ > Ag_3_PO_4_/BiPO_4_ > Ag_3_PO_4_ > BiPO_4_. The result indicates that the CQDs/Ag_3_PO_4_/BiPO_4_ composite has the highest photocatalytic activity. It should be noted out that photosensitized degradation of RhB could occur in the present photocatalytic system. However, the photosensitization effect is not the dominant degradation mechanism since Ag_3_PO_4_ based composite photocatalysts have also been demonstrated to exhibit pronounced degradation of colorless phenol [65].

The first order kinetic rate of the dye degradation photocatalyzed by the samples can be evaluated by Equation (2) [66]:Ln(*C*_t_/*C*_0_) = −*kt*(2)
where *C*_0_, *C*_t_, *k*, and *t* is the initial concentration of RhB, apparent concentration of RhB after degradation, kinetic rate constant, and irradiation time, respectively. Figure 8b shows the plots of Ln(*C*_t_/*C*_0_) vs. *t*. The rate constant (*k*) for the photocatalysts is found to be *k*_BiPO4_ = 0.00261, *k*_Ag3PO4_ = 0.02853, *k*_Ag3PO4/BiPO4_ = 0.04489, and *k*_CQDs/Ag3PO4/BiPO4_ = 0.08259 min^−1^. The result further indicates that the CQDs/Ag_3_PO_4_/BiPO_4_ composite exhibits a photocatalytic activity for the degradation of RhB 31.6, 2.9, and 1.8 times higher than that of BiPO_4_, Ag_3_PO_4_ and Ag_3_PO_4_/BiPO_4_, respectively. We compare the photodegradation performance of CQDs/Ag_3_PO_4_/BiPO_4_ with that of other typical composite photocatalysts, as shown in Table 2. It is seen that the CQDs/Ag_3_PO_4_/BiPO_4_ composite photocatalyst prepared in this work manifests a photodegradation performance superior to most of other photocatalysts.

The stability and reusability of the CQDs/Ag_3_PO_4_/BiPO_4_ photocatalyst was performed by repeating the experiments for the degradation of the RhB dye under simulated sunlight irradiation, as shown in Figure 9. It is seen that, after five cycles, no obvious decrease in the dye degradation is observed, which indicates that the CQDs/Ag_3_PO_4_/BiPO_4_ photocatalyst has a high stability and maintains a high photocatalytic activity for the degradation of RhB.

### 3.7. Photocatalytic Mechanism

Figure 10a schematically shows the assembly structure of the CQDs/Ag_3_PO_4_/BiPO_4_ composite with Ag_3_PO_4_ fine particles and CQDs homogenously decorated on the surface of rice-like BiPO_4_ granules. A possible photocatalytic mechanism of the CQDs/Ag_3_PO_4_/BiPO_4_ composite toward the degradation of RhB under simulated sunlight irradiation is schematically depicted in Figure 10b. The conduction band (CB) and valence band (VB) potentials of BiPO_4_ and Ag_3_PO_4_ can be calculated by using Equations (3) and (4) [74,75]:*E*_CB_ = *X* − *E*^e^ − 0.5*E*_g_(3)
*E*_VB_ = *X* − *E*^e^ + 0.5*E*_g_(4)
where *E*^e^ is 4.5 eV, being the free electron energy on the hydrogen scale. *X*_Ag3PO4_ and *X*_BiPO4_ are estimated as 5.959 and 6.633 eV, respectively, according to Equations (5) and (6):
(5)$$X(Ag_{3}PO_{4})=\sqrt[{8}]{X{\left(Ag\right)}^{3}X(P)X{\left(O\right)}^{4}}$$
(6)$$X(BiPO_{4})=\sqrt[{6}]{X(Bi)X(P)X{\left(O\right)}^{4}}$$
where *X*(Ag) = 4.44, *X*(P) = 5.62, *X*(Bi) = 4.69, and *X*(O) = 7.54 eV. The CB potentials of BiPO_4_ and Ag_3_PO_4_ are estimated as −0.127, and +0.222 V, respectively, and their corresponding VB potentials are +4.434, and +2.691 V. For the Ag_3_PO_4_/BiPO_4_ composite, the energy band of Ag_3_PO_4_ is completely located within the energy band of BiPO_4_. Therefore, the Ag_3_PO_4_/BiPO_4_ composite obeys the type-I band alignment. When CQDs are introduced to the Ag_3_PO_4_/BiPO_4_ composite, it promotes the charge transfer between the two kinds of semiconductors. When the CQDs/Ag_3_PO_4_/BiPO_4_ photocatalyst is irradiated by simulated sunlight, the electron transition occurs from the VB to the CB of Ag_3_PO_4_, thus producing electron-hole pairs. Subsequently, the holes in the VB of Ag_3_PO_4_ react with the RhB dye to form degradation products. Simultaneously, CQDs can be also excited by absorbing visible light, i.e., the π electrons or σ electrons are excited to the lowest unoccupied molecular orbital (LUMO) [76,77]. The excited CQDs can be acted as excellent electron donors and acceptors. However, BiPO_4_ could not be photoexcited to generate electron-hole pairs under simulated sunlight irradiation due to its large bandgap energy (4.561 eV). Consequently, the CB electrons in Ag_3_PO_4_ will transfer to CQDs (π or σ orbitals), and the photoexcited electrons in CQDs will transfer to the CB of BiPO_4_. Due to this interesting electron transfer process, the recombination of the photoexchited electron-hole pairs in Ag_3_PO_4_ are efficiently suppressed. Furthermore, the up-conversion photoluminescence emitted from CQDs could excite Ag_3_PO_4_ to generate additional electron-hole pairs. The photoexcited electrons in the LUMO of CQDs and those relaxed to the CB of BiPO_4_ react with oxygen in the photocatalytic system to form superoxide (•O_2_^−^) radicals. The produced •O_2_^−^ radicals react with dye molecules adsorbed on the surface of the photocatalyst to produce degradation products.

## 4. Conclusions

A simple hydrothermal method has been used to synthesize the CQDs/Ag_3_PO_4_/BiPO_4_ heterostructure photocatalyst. The carbon quantum dots are anchored at the interfaces between Ag_3_PO_4_ and BiPO_4_, thus forming the CQDs/Ag_3_PO_4_/BiPO_4_ three-phase junction structure. The three-phase junction structure results in an efficient charge separation and utilization, high light absorption capacity and low photoluminescence intensity. The CQDs/Ag_3_PO_4_/BiPO_4_ composite exhibits significantly enhanced photocatalytic activity for the degradation of RhB, which can be explained as the result of efficient charge separation and increased visible-light absorption.

## Figures and Tables

**Figure 1 micromachines-10-00557-f001:**
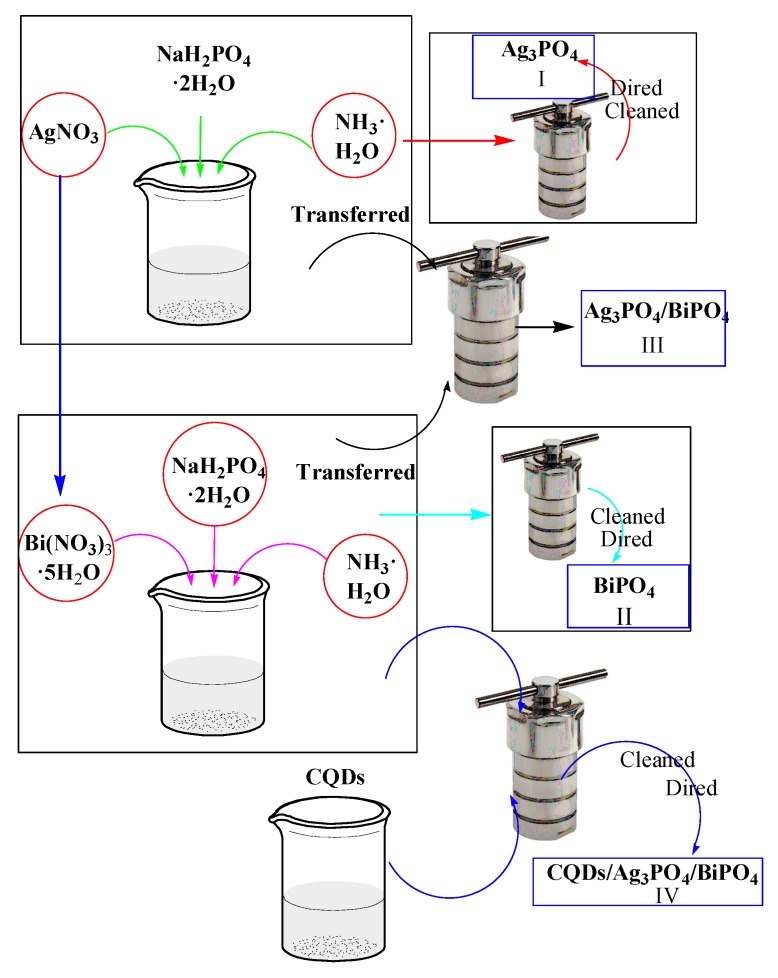
Chemical route for the preparation of (I) Ag_3_PO_4_, (II) BiPO_4_, (III) Ag_3_PO_4_/BiPO_4_, and (IV) CQDs/Ag_3_PO_4_/BiPO_4_.

**Figure 2 micromachines-10-00557-f002:**
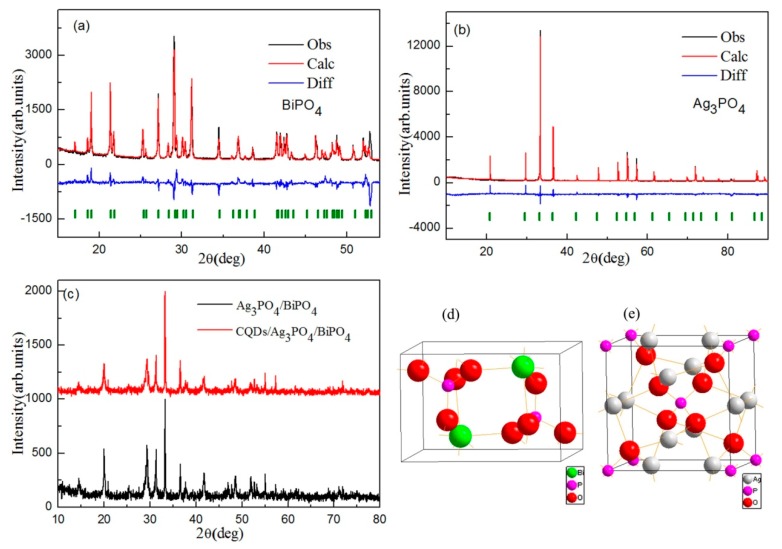
XRD patterns of (**a**) BiPO_4_, (**b**) Ag_3_PO_4_, (**c**) Ag_3_PO_4_/BiPO_4_ and CQDs/Ag_3_PO_4_/BiPO_4_, and crystal structures of (**d**) BiPO_4_ and (**e**) Ag_3_PO_4_.

**Figure 3 micromachines-10-00557-f003:**
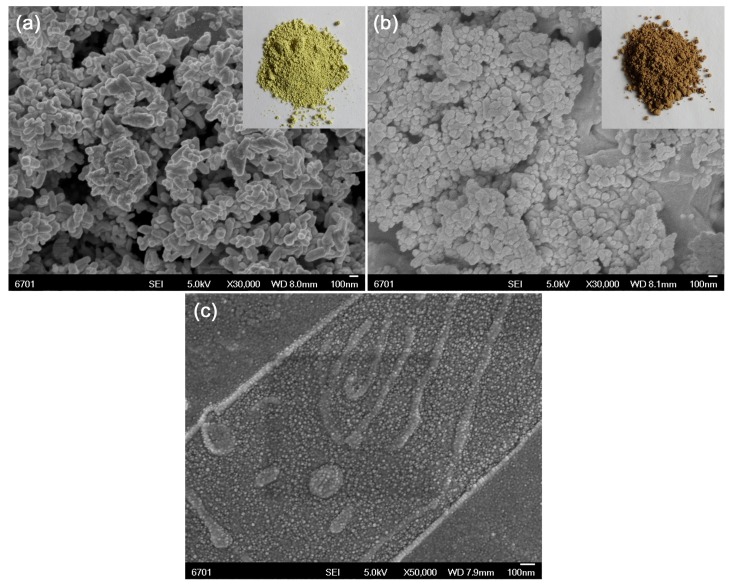
SEM images of (**a**) Ag_3_PO_4_/BiPO_4_, (**b**) CQDs/Ag_3_PO_4_/BiPO_4_, and (**c**) pure CQDs. The insets represent the real photos of Ag_3_PO_4_/BiPO_4_ and CQDs/Ag_3_PO_4_/BiPO_4_.

**Figure 4 micromachines-10-00557-f004:**
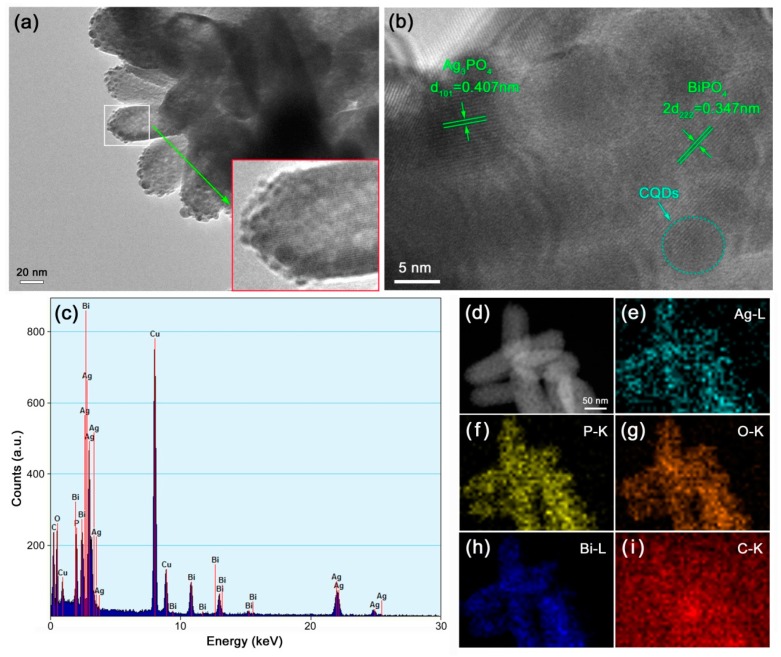
TEM image (**a**), HRTEM image (**b**), EDS spectrum (**c**), DF-STEM image (**d**), and elemental mapping images (**e**–**i**) of the CQDs/Ag_3_PO_4_/BiPO_4_ composite.

**Figure 5 micromachines-10-00557-f005:**
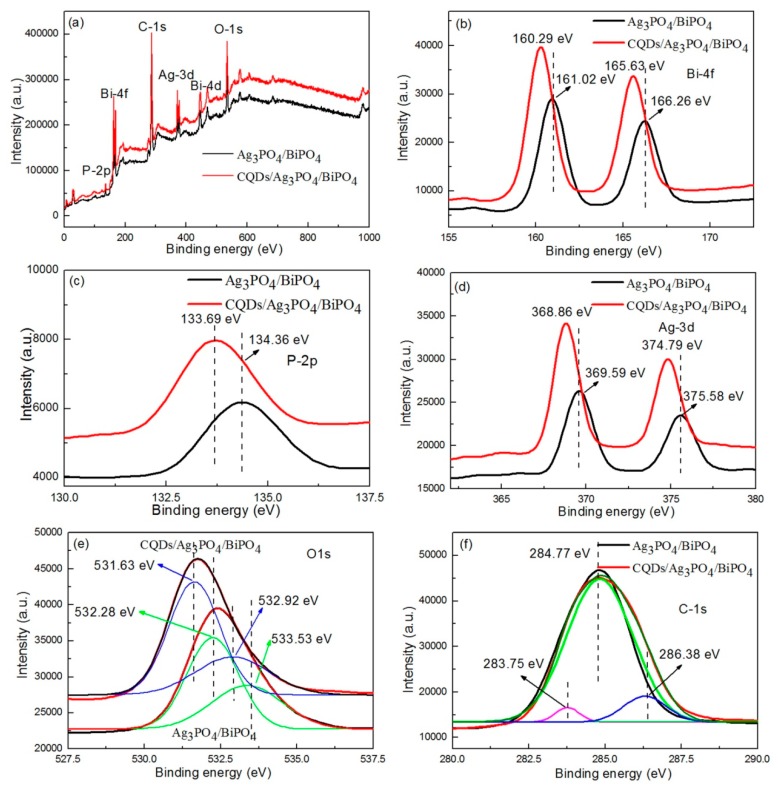
XPS survey scan spectra (**a**), Bi 4f spectra (**b**), P 2p spectra (**c**), Ag 3d spectra (**d**), O 1s spectra (**e**), and C 1s spectra (**f**) of the Ag_3_PO_4_/BiPO_4_ and CQDs/Ag_3_PO_4_/BiPO_4_ composites.

**Figure 6 micromachines-10-00557-f006:**
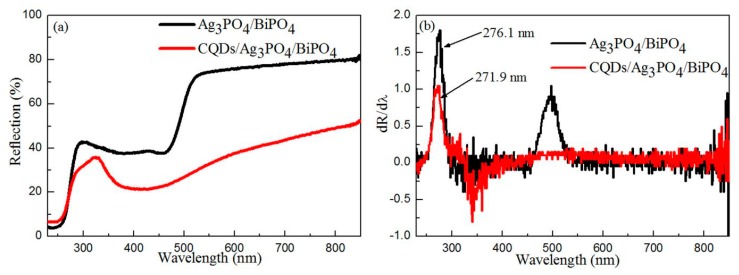
UV–VIS diffuse reflectance spectra (**a**) and the corresponding first derivative curves of the UV–VIS diffuse reflectance spectra (**b**) of the Ag_3_PO_4_/BiPO_4_ and CQDs/Ag_3_PO_4_/BiPO_4_ composites.

**Figure 7 micromachines-10-00557-f007:**
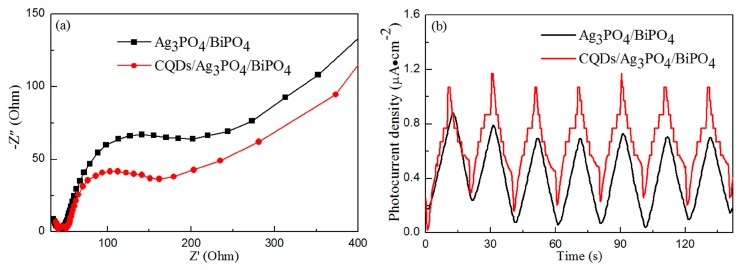
EIS spectra (**a**) and photocurrent response curves (**b**) of the Ag_3_PO_4_/BiPO_4_ and CQDs/Ag_3_PO_4_/BiPO_4_ composites.

**Figure 8 micromachines-10-00557-f008:**
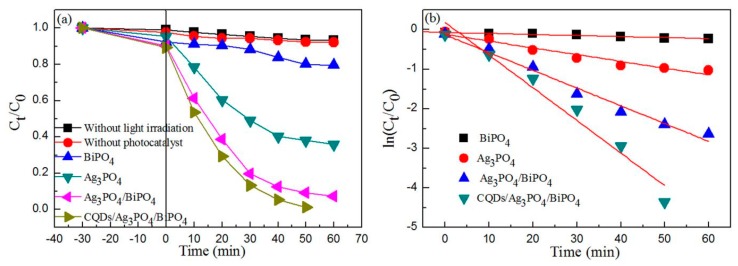
(**a**) Time-dependent photocatalytic degradation of RhB dye over the BiPO_4_, Ag_3_PO_4_, Ag_3_PO_4_/BiPO_4_, and CQDs/Ag_3_PO_4_/BiPO_4_ photocatalysts under simulated sunlight irradiation. (**b**) Plots of Ln(*C*/*C*_0_) vs. irradiation time for the samples.

**Figure 9 micromachines-10-00557-f009:**
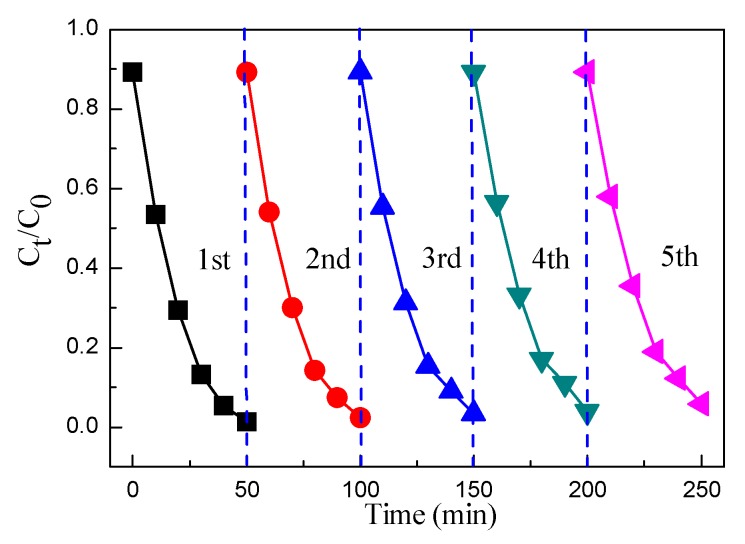
Recyclability of the CQDs/Ag_3_PO_4_/BiPO_4_ photocatalyst for the photocatalytic degradation of RhB under simulated sunlight irradiation.

**Figure 10 micromachines-10-00557-f010:**
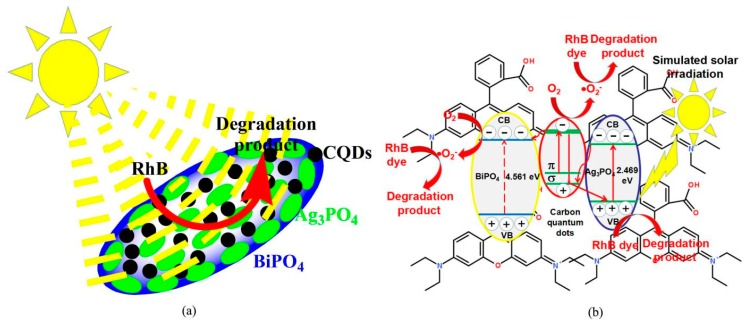
Schematic illustration of the assembly structure (**a**) and a possible photodegradation mechanism (**b**) of the CQDs/Ag_3_PO_4_/BiPO_4_ composite.

**Table 1 micromachines-10-00557-t001:** Color coordinates and *E*_g_ values of Ag_3_PO_4_/BiPO_4_ and CQDs/Ag_3_PO_4_/BiPO_4_.

Sample	Color Coordinates						*E*_g_ of Ag_3_PO_4_ (eV)	*E*_g_ of BiPO_4_ (eV)
	*L**	*a**	*b**	*c**	*H* ^o^	*E*_CIE_*		
Ag_3_PO_4_/BiPO_4_	87.110	−3.666	30.235	30.456	−83.087	92.281	2.469	4.491
CQDs/Ag_3_PO_4_/BiPO_4_	63.817	4.286	15.421	16.006	74.468	65.794	-	4.561

**Table 2 micromachines-10-00557-t002:** Comparison of the photocatalytic performance of CQDs/Ag_3_PO_4_/BiPO_4_ with that of previously reported Ag_3_PO_4_-based composite photocatalysts toward the degradation of RhB.

Samples	Light Source	*C*_photocatalyst_ (g L^−1^)	*C*_RhB_ (mg L^−1^)	Irradiation Time (min)	D%	Reference
CQDs/Ag_3_PO_4_/BiPO_4_	200 W Xe lamp	1	5	50	98.7	This work
20wt%Ag_3_PO_4_/Bi_2_WO_6_	200 W Xe lamp	0.5	5	120	94	[9]
10% Bi_4_Ti_3_O_12_/Ag_3_PO_4_	200 W Xe lamp	0.2	5	30	99.5	[6]
Ag-Ag_3_PO_4_	30 W fluorescent light lamp (λ ≥ 420 nm)	0.75	10	60	70	[67]
Fe_3_O_4_/ZnO/Ag_3_PO_4_	50 W LED lamp	0.4	12 (10^−5^ mol/L)	100	75	[68]
15 wt% Ag_3_PO_4_-Bi_2_MoO_6_	300 W Xe lamp with a 400-nm cutoff filter	1	10	100	39	[69]
Ag_3_PO_4_-ZnO (1:40)	300 W Xe lamp with a 400-nm cutoff filter	0.67	12 (10^−5^ mol/L)	30	93	[70]
Ag_3_PO_4_	15 W four fluorescent lamp	0.3	15	60	75	[71]
AgI/BiPO_4_	500 W Xe lamp with a 420-nm cutoff filter	1.67	10	60	92.2	[72]
Ag_2_S/CQDs/CuBi_2_O_4_	200 W Xe lamp	1	5	60	99.3	[45]
BiPO_4_/Ag/Ag_3_PO_4_	150 W Xe lamp with a 420-nm cutoff filter	0.1	20	120	65	[73]

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
