# Peer review of "Construction of a CQDs/Ag3PO4/BiPO4 Heterostructure Photocatalyst with Enhanced Photocatalytic Degradation of Rhodamine B under Simulated Solar Irradiation"

_micromachines, 2019, doi:10.3390/mi10090557_

Round 1

Reviewer 1 Report

Authors reported ternary photocatalysts of CQDS/Ag3PO4/ BiPO4 heterostructures, employed for degradation RhB dye. This work is interesting. However, authors need to address the following concerns before accept for publication.

(i)                  Authors stated that CQDs in ternary heterostructure. However, from HRTEM, CQDS is not clear in heterostructure. I recommended to authors, please include the pure CQDs HRTEM and size distribution of them. Otherwise, edit the title as Carbon nanoparticles (CNPs)/Ag3PO4/BiPO4. Authors fails to give proper evidence for QDs of carbon. Need more attention for rewrite the title and manuscript content.

(ii)                In introduction and results and discussion parts needed to improve and added these relevant articles and compare the photoactivity of ternary system. Highlight the importance and role of Carbon NPS in heterostructure.

(1)    Ceramics International 44 (10), 11783-11791; (2) Applied Surface Science 447, 740-756; (3) ACS Omega 3 (7), 7587-7602; (4) Scientific reports 8 (1), 4194.

(iii)               Why photocurrent curves have not obvious pattern. I guess leaching of materials during the test. Please reconduct the experiment again if possible. Please check the Y-axis scale.

(iv)              Please describe the working electrode preparation and add the procedure of electrochemical analysis  in characterization part.

(v)                What electrolyte is used for photocurrent test? Please specify in details.

(vi)              In EDX spectra, why Cu peaks appeared?

(vii)               In photocatalytic mechanism, it showing that carbon transferring the electrons to BiPO4. Please recheck the mechanism. Please check the mechanism and rewrite with respect of their band alignments.

Author Response

Response to Reviewer#1

Authors reported ternary photocatalysts of CQDS/Ag3PO4/ BiPO4 heterostructures, employed for degradation RhB dye. This work is interesting. However, authors need to address the following concerns before accept for publication.

(i)                  Authors stated that CQDs in ternary heterostructure. However, from HRTEM, CQDS is not clear in heterostructure. I recommended to authors, please include the pure CQDs HRTEM and size distribution of them. Otherwise, edit the title as Carbon nanoparticles (CNPs)/Ag3PO4/BiPO4. Authors fails to give proper evidence for QDs of carbon. Need more attention for rewrite the title and manuscript content.

Answer: We thank very much the reviewer for carefully reading out manuscript and giving us valuable comments and suggestions. This comment is very good. According to the reviewer’s suggestion, we have provided the SEM image of pure CQDs (Fig. 3c), from which it is seen that the prepared carbon particles have a size distribution of 7-10 nm.

(ii)                In introduction and results and discussion parts needed to improve and added these relevant articles and compare the photoactivity of ternary system. Highlight the importance and role of Carbon NPS in heterostructure.

(1)    Ceramics International 44 (10), 11783-11791; (2) Applied Surface Science 447, 740-756; (3) ACS Omega 3 (7), 7587-7602; (4) Scientific reports 8 (1), 4194.

Answer: Thanks. We have cited the good papers mentioned by the reviewer.

(iii)               Why photocurrent curves have not obvious pattern. I guess leaching of materials during the test. Please reconduct the experiment again if possible. Please check the Y-axis scale.

Answer: Thank the reviewer for giving us this valuable comment. We agree fully with the reviewer’s comment that the photocurrent curves are not good. However, re-measuring photocurrent curves will take a long time because the universities in China are now on holiday. We have carefully checked the Y-axis scale and corrected.

(iv)              Please describe the working electrode preparation and add the procedure of electrochemical analysis  in characterization part.

Answer: Thank the reviewer for giving us this good suggestion. According to the reviewer’s suggestion, we have added the description on the working electrode preparation and electrochemical analysis.

(v)                What electrolyte is used for photocurrent test? Please specify in details.

Answer: The used electrolyte was 0.1 mol L1 Na2SO4 aqueous solution. We have added this information in 2.5 Sample characterization.

(vi)              In EDX spectra, why Cu peaks appeared?

Answer: Thanks. Additional Cu signal observed on the EDS spectrum can be ascribed to the TEM microgrid holder. We have added this information in page 6.

(vii)               In photocatalytic mechanism, it showing that carbon transferring the electrons to BiPO4. Please recheck the mechanism. Please check the mechanism and rewrite with respect of their band alignments.

Answer: Thank the reviewer for giving this comment. We have carefully checked the photocatalytic mechanism and made minor amendment. It is noted that Ag3PO4 and CQDs can be photoexcited, whereas BiPO4 can not be photoexcited under simulated sunlight irradiation due to its large bandgap energy (4.561 eV). The excited CQDs can be acted as excellent electron donors and acceptors (here we have cited the related papers). As a result, the CB electrons in Ag3PO4 will transfer to CQDs (p or s orbitals), and the photoexcited electrons in CQDs will transfer to the CB of BiPO4. This electron transfer process is reasonable explained and efficiently suppresses the recombination of the photoexchited electron-hole pairs in Ag3PO4. This is the dominant reason for the photocatalytic enhanced mechanism.

Reviewer 2 Report

This paper reports the Construction of CQDs/Ag3PO4/BiPO4 heterostructure photocatalyst with enhanced photocatalytic degradation of rhodamine B under simulated solar irradiation. The analysis of data and discussion about the material characterization and catalytic activity is so poor. Therefore, I suggest the authors to address all the following questions before possible for publication.

(1)    Photo-current measurements should be added and correlated the result with photocatalysis mechanism. These further measurements are essential, to know the charge recombination process.

(2)    In the present study, the catalytic activity of the composite was evaluated by the degradation of dye: To confirm the degree of degradation, the TOC analysis should be carried out.

(3)    To rule out the photosensitization effect under light irradiation, better to evaluate the photocatalytic performance of nanocomposites toward the colorless phenol.

(4)    Structural stability is important factor for photocatalytic materials, so, the authors should provide XRD and XPS results to confirm the stability after recycling measurements.

(5)    Is the dye oxidized completely? Further experimental evidences required to know the end products like LCMS or HPLC

(6)    There are many typo and grammatical mistakes that should be revised carefully.

Author Response

Response to Reviewer#2

This paper reports the Construction of CQDs/Ag3PO4/BiPO4 heterostructure photocatalyst with enhanced photocatalytic degradation of rhodamine B under simulated solar irradiation. The analysis of data and discussion about the material characterization and catalytic activity is so poor. Therefore, I suggest the authors to address all the following questions before possible for publication.

(1)    Photo-current measurements should be added and correlated the result with photocatalysis mechanism. These further measurements are essential, to know the charge recombination process.

Answer: We have added the photo-current data (Fig. 10b) and made the corresponding analysis.

(2)    In the present study, the catalytic activity of the composite was evaluated by the degradation of dye: To confirm the degree of degradation, the TOC analysis should be carried out.

Answer: It should be noted that Ag3PO4 is a famous photocatalyst that can efficiently degrade organic dye. TOC analysis is generally necessary for a new photocatalyst, but it is not always indispensable for a widely studied photocatalyst like Ag3PO4. This is why many publications related to photocatalysis do not include the TOC data. Also, our lab has no total organic carbon analyzer available for the TOC analysis.

(3)    To rule out the photosensitization effect under light irradiation, better to evaluate the photocatalytic performance of nanocomposites toward the colorless phenol.

Answer: Regarding this comment, we have discussed the photosensitization effect based on previous reported results.

(4)    Structural stability is important factor for photocatalytic materials, so, the authors should provide XRD and XPS results to confirm the stability after recycling measurements.

Answer: The recycling photocatalytic experiments clearly suggest that the CQDs/Ag3PO4/BiPO4 photocatalyst has a high stability and reusability for the degradation of RhB (Fig. 12). To provide additional XRD and XPS data will take a long time and delay the publication of the paper, because the universities in China are now on holiday.

(5)    Is the dye oxidized completely? Further experimental evidences required to know the end products like LCMS or HPLC

Answer: This comment is similar to comment (2). The end products analysis is not always indispensable for a famous photocatalyst like Ag3PO4. And moreover, the precise determination of end products is rather complicated. The instruments that can be used for the analysis of end products (e.g. chromatograph and mass spectrometers) are absent in our lab.

(7)    There are many typo and grammatical mistakes that should be revised carefully.

Answer: We have carefully checked and revised typo and grammatical mistakes.

Reviewer 3 Report

I think the paper is interesting and technically correct.

I have some concerns with the optical characterization part:

minor concern Formula (1) at line 231. This particular relation holds if Eg is in eV and lambda in nm. It should be mentioned, either a reader assumes the quantities ar in the S.I. system and the formula does not hold. Eg(eV)=1240/(lambda (nm)) eV major concerns is about the the PL spectra in Figg. 7 and 8. What is the laser pump power incident on the samples? If too low (as it could be using a monochromatized lamp of a spectrofluorimeter which can deliver few microwatts of pump power over the sample) many PL "ghosts" could appear and the literature of is full of such issues. Look for example here: https://www.edinst.com/wp-content/uploads/2016/07/AN_P35-Charge-carrier-recombination-v.2.pdf or Jiang, X. et al. Characterization of Oxygen Vacancy Associates within Hydrogenated TiO2: A Positron Annihilation Study. J. Phys. Chem. C 116, 22619–22624 (2012) or Rex, R. E., Knorr, F. J. & McHale, J. L. Comment on ‘Characterization of Oxygen Vacancy Associates within Hydrogenated TiO2: A Positron Annihilation Study’. J. Phys. Chem. C 117, 7949–7951 (2013). Please explicitates this value and please assess better if the PL spectra are those expected  from Ag3PO4 and BiPO4 at room temperature. 

For what concerns the general message of the paper, a comparison with other catalytic systems (TiO2? Other?) could help in reinforcing the claims about "enhancement" in the photocatalytic properties of such heterostructures. 

Author Response

Response to Reviewer#3

I think the paper is interesting and technically correct.

I have some concerns with the optical characterization part:

minor concern Formula (1) at line 231. This particular relation holds if Eg is in eV and lambda in nm. It should be mentioned, either a reader assumes the quantities ar in the S.I. system and the formula does not hold. Eg(eV)=1240/(lambda (nm)) eV major concerns is about the the PL spectra in Fig. 7 and 8. What is the laser pump power incident on the samples? If too low (as it could be using a monochromatized lamp of a spectrofluorimeter which can deliver few microwatts of pump power over the sample) many PL "ghosts" could appear and the literature of is full of such issues. Look for example here: https://www.edinst.com/wp-content/uploads/2016/07/AN_P35-Charge-carrier-recombination-v.2.pdf or Jiang, X. et al. Characterization of Oxygen Vacancy Associates within Hydrogenated TiO2: A Positron Annihilation Study. J. Phys. Chem. C 116, 22619–22624 (2012) or Rex, R. E., Knorr, F. J. & McHale, J. L. Comment on ‘Characterization of Oxygen Vacancy Associates within Hydrogenated TiO2: A Positron Annihilation Study’. J. Phys. Chem. C 117, 7949–7951 (2013). Please explicitates this value and please assess better if the PL spectra are those expected from Ag3PO4 and BiPO4 at room temperature. 

Response: We thank very much the reviewer for carefully reading our manuscript and giving us valuable suggestions.

1) The relevant units of formula (1) have been added in Eq. (1).

2) The laser pump power has been added in 2.5 Sample characterization.

3) We used xenon lamps to monitor the fluorescence properties of as-prepared samples. According to the reviewer’s suggestion, we have added the following sentence and cited the papers mentioned by the reviewer.

“However, it can not be ruled out that the two emission peaks at 540 and 560 nm are possibly artifacts due to the low power xenon lamp used as the excitation source.”

For what concerns the general message of the paper, a comparison with other catalytic systems (TiO2? Other?) could help in reinforcing the claims about "enhancement" in the photocatalytic properties of such heterostructures. 

Response: We added the following contents to support this view.

“Compared to Ag2S/CQDs/CuBi2O4 [45], the CQDs/Ag3PO4/BiPO4 composite requires only 50 minutes to completely degrade RhB dye under the same conditions, indicating that the CQDs/Ag3PO4/BiPO4 composite exhibits a high photocatalytic activity for the degradation of RhB dye.”

Round 2

Reviewer 1 Report

Authors addressed all concerns raised by reviewer and improved the quality of the manuscript.

Author Response

We thank very much the reviewer for kindly reviewing our revised manuscript.

Reviewer 3 Report

As from the authors' report, they cannot exclude that the photoluminescence emissions are artifacts. These data (photoluminescence) then need to be excluded if they cannot re-do under laser (mW) excitation. Please modify the chapter Optical Properties. 

For what concerns the new sentence “Compared to Ag2S/CQDs/CuBi2O4 [45], the CQDs/Ag3PO4/BiPO4 composite requires only 50 minutes to completely degrade RhB dye under the same conditions, indicating that the CQDs/Ag3PO4/BiPO4 composite exhibits a high photocatalytic activity for the degradation of RhB dye.” there is not a comparison with another assessed catalytic system. "Only 50 ' " is a justified comparison if authors can say how does it takes to degrade RhB dye with other catalytic system. I suggest the authors to add such a comparison.

Author Response

As from the authors' report, they cannot exclude that the photoluminescence emissions are artifacts. These data (photoluminescence) then need to be excluded if they cannot re-do under laser (mW) excitation. Please modify the chapter Optical Properties.

Response: Thank you very much for your valuable advice. In the preliminary work, we have carried on the experiment of obtaining fluorescence data by laser light source (References [27] and [67]). Due to we need to make an appointment with other research groups for testing, the experiment of using high power laser light source to test the photoluminescence properties is temporarily unable to be performed during the holidays. In the follow-up research work, we will study the influence of different light sources and different environments on the photoluminescence properties of as-prepared samples on the basis of your valuable ideas. To characterize the photoluminescence properties of Ag3PO4, BiPO4, Ag3PO4/BiPO4 and (CQDs)/Ag3PO4/BiPO4 is icing on the cake. Thus, according to the reviewer’s suggestion, we have deleted the relevant characterization and content.

For what concerns the new sentence “Compared to Ag2S/CQDs/CuBi2O4 [45], the CQDs/Ag3PO4/BiPO4 composite requires only 50 minutes to completely degrade RhB dye under the same conditions, indicating that the CQDs/Ag3PO4/BiPO4 composite exhibits a high photocatalytic activity for the degradation of RhB dye.” there is not a comparison with another assessed catalytic system. "Only 50 ' " is a justified comparison if authors can say how does it takes to degrade RhB dye with other catalytic system. I suggest the authors to add such a comparison.

Response: According to the reviewer’s good suggestion, we have provided a Table (Table 2) to compare the photodegradation performance of the CQDs/Ag3PO4/BiPO4 composite prepared in this work with that of previously reported composite photocatalysts. Relevant content has been added in Page 11, lines 289-296.

The following reference was added to the text to support our view.

“9. Zheng, C.X.; Yang, H. Assembly of Ag3PO4 nanoparticles on rose flower-like Bi2WO6 hierarchical architectures for achieving high photocatalytic performance. J. Mater. Sci.-Mater. Electron. 2018, 29, 9291–9300.

Zheng, C.X.; Yang, H.; Cui, Z.M.; Zhang, H.M.; Wang, X.X. A novel Bi4Ti3O12/Ag3PO4 heterojunction photocatalyst with enhanced photocatalytic performance. Nanoscale Res. Lett. 2017, 12, 608. Lei, Y.; Wang, G.; Guo, P.; Song, H.C. Silver phosphate based plasmonic photocatalyst: highly active visible-light-enhanced photocatalytic property and photosensitized degradation of pollutants. Funct. Mater. Lett. 2012, 5, 1250047. Shekofteh-Gohari, M.; Habibi-Yangjeh, A. Photosensitization of Fe3O4/ZnO by AgBr and Ag3PO4 to fabricate novel magnetically recoverable nanocomposites with significantly enhanced photocatalytic activity under visible-light irradiation. Ceram. Int. 2016, 42, 15224–15234. Du, X.; Wan, J.; Jia, J.; Pan, C.; Hu, X.; Fan, J.; Liu, E. Photocatalystic degradation of RhB over highly visible-light-active Ag3PO4-Bi2MoO6 heterojunction using H2O2 electron capturer. Mater. Design, 2017,119, 113–123. Dong, C.; Wu, K.L.; Li, M.R.; Liu, L.; Wei, X.W. Synthesis of Ag3PO4–ZnO nanorod composites with high visible-light photocatalytic activity. Catal. Commun. 2014, 46, 32–35. Vu,T.A.; Dao, C.D.; Hoang, T. T.T.; Nguyen, K.T.; Le, G.H.; Dang, P.T.; Tran, H.T.K.; Nguyen, T.V. Highly photocatalytic activity of novel nano-sized Ag3PO4 for Rhodamine B degradation under visible light irradiation. Mater. Lett. 2013, 92, 57–60. Ye, H.; Lin, H.; Cao, J.; Chen, S.; Chen, Y. Enhanced visible light photocatalytic activity and mechanism of BiPO4 nanorods modified with AgI nanoparticles. J. Mol. Catal. A-Chem. 2015, 397, 85–92. Mohaghegh, N.; Rahimi, E. BiPO4 photocatalyst employing synergistic action of Ag/Ag3PO4 nanostructure and graphene nanosheets. Solid State Sci. 2016, 56, 10-15.”

Round 3

Reviewer 3 Report

I think the paper now is worth to be published eventhough a critical assessment of the differences (improvements vs weak points) of this photocatalytic system vs thos presented in the new table 2 could have been informative.